# Retrieval of refractive index and water content for the coating materials of aged black carbon aerosol based on optical properties: a theoretical analysis

Jia Liu[a,b,c], Cancan Zhu[a,b,c], Donghui Zhou[a,b,c], Jinbao Han[a,b,c]

[a] Non-destructive Testing Laboratory, School of Quality and Technical Supervision, Hebei University, Baoding 071002, China.
[b] Engineering Research Center of Zero-carbon Energy Buildings and Measurement Techniques, Ministry of Education, Hebei University, Baoding 071002, China.
[c] Hebei Key Laboratory of Energy Metering and Safety Testing Technology, Hebei University, Baoding 071002, China.

**Abstract.** Water content in the coatings of aged black carbon (BC) aerosol can be reflected through complex refractive index. In this study, the retrieval of refractive index and water content for the non-absorbing coatings of BC aerosol during the hygroscopic growth (RH=0-95%) based on scattering and absorption properties is theoretically investigated. Optical properties of morphologically realistic fractal BC aerosols are simulated using the Multiple Sphere *T*-matrix method (MSTM), the optical equivalent refractive index of coating material is retrieved based on the Mie theory, and the water content in coatings is further retrieved using effective medium theory. Results show that the scattering property performs best in retrieving refractive index and water content. The retrieval errors of the refractive index of heavily-aged BC aerosols are less than 10% at high RHs, while partially-coated and thinly-coated BC have larger errors. The regularity of retrieved water content is similar to that of refractive index retrieve, and the water content retrieved errors range from 2% to 63% for heavily-coated BC. This study provides a helpful optical method to obtain the water content of BC coatings.

## 1 Introduction

Black carbon (BC) aerosols are ubiquitous in the Earth's atmosphere and directly lead to global and regional warming by scattering and absorbing solar radiation(Zhang et al., 2019b; Zhang et al., 2020). Since BC is mainly produced by incomplete combustion of fossil fuel and biomass, various microphysical characteristics determine the diversity of scattering and absorption properties, which further bring huge uncertainty in the radiative and climate effects of BC (Zhang et al., 2019a). Fresh bare BC will be coated by inorganic salts or organics during aging process such as condensation and collision in the atmosphere, and hydrophobic BC aerosol becomes hydrophilic. Zhang et al. (2023) studied the collapse of particle soot structure and changes in coating composition during long-distance transport. The results showed that when the relative humidity (RH) is between 60% and 90%, it is conducive to forming secondary aerosol coatings on soot particles and facilitates the transition of soot from a partially coated state to an embedded state. Soot-aggregate restructuring is a complex

phenomenon influenced by various factors, including the physical and chemical properties of the coating materials and the environmental conditions to which the soot is exposed. Soot compaction is mainly influenced by internal mixing mechanisms. Soot is partially compacted before full coating and typically becomes fully compacted after a fivefold increase in volume, regardless of the coating material (Corbin et al., 2023; Sipkens and Corbin, 2024). Coating materials with different complex refractive indices produce different "lensing effect" or "sunglass effect" (Liu et al., 2021; Feng et al., 2021). In addition, the optical properties of coated BC are significantly different from those of bare BC due to the morphological changes of fractal structure, thus increasing the uncertainty of radiative effect (Luo et al., 2018b; Fierce et al., 2016; Li et al., 2024b; Wang et al., 2021b; Wu et al., 2017; Pang et al., 2023; Mishchenko et al., 1995). Therefore, the determination of complex refractive indices (CRIs) of coated BC and even only the coating material is essential for field and laboratory observations.

Refractive index of aerosols, $m=n+ki$, can be determined from the scattering and absorption properties, the real part is related to the former and the imaginary part is related to the latter. Tan et al. (2013) conducted field observation with a hygroscopic tandem differential mobility analyzer (H-TDMA) under high humidity (about 90%) at Pearl River Delta, which also has a high BC concentration. In order to characterize the aerosol liquid water contents (ALWC) at North China Plain, Kuang et al. (2018) employed a three-wavelength humidified nephelometer system to measure optical properties at different RHs and they stressed that the measured ALWC was in good agreement with the calculated results of thermodynamic model. Zhou et al. (2020) measured the scattering, absorption, extinction coefficients and SSA at 532 nm using a humidifier cavity-enhanced albedometer (H-CEA) in the laboratory, the relative humidity of H-CEA could increase from 10% to 88%. With the assistance of a self-developed cavity-enhanced albedometer, (Zhao et al., 2014; Xu et al., 2016) measured the extinction coefficient, scattering coefficient, absorption coefficient and single scattering albedo (SSA) for atmospheric aerosols at Jing-Jin-Ji Area. The effective CRI of aerosols is retrieved based on the Mie theory of homogeneous sphere by using the optical properties and volume mixing. The real part of CRIs is about 1.38 ~ 1.44, and the imaginary part is about 0.008 ~ 0.04. Zhao et al. (2019) combined a differential mobility analyzer (DMA) and a single-particle soot photometer (SP2) to characterize the scattering properties of size-resolved ambient aerosol at a suburban site, the real part of CRIs retrieved using Mie calculation was 1.34-1.56 and increased slowly with the aerosol diameter. At four different sites in China, Zhao et al. (2021) revealed that the real part of CRIs ranged from 1.36 to 1.78 and increased with the mass ratio of organics. Radney and Zangmeister (2018) compared two aerosol CRI retrieval methods based on Mie scattering theory, the first method employed measurements of optical properties of size-selected particles while the second method employed measurements of both optical properties and particle size distributions, they recommended the application of these methods in laboratory and field observations respectively. Through the combination of a novel broadband cavity enhanced spectroscopy and a DMA, the aging process of pinene and xylene and the production of secondary organic aerosols (SOA) in an oxidation flow reactor were investigated by He et al. (2018), extinction properties were used to retrieve CRIs, results showed that SOAs are not absorbing in visible range while the real part of CRIs depends slightly on wavelength. With special attention to coatings of BC, Xu et al. (2018) investigated the optical properties of aerosols in field observations using an albedometer, based on core-

shell Mie theory for coated BC, the imaginary part of CRIs was retrieved to be 0.004-0.008. Through an instrument setup consisting of aethalometer, nephelometer, aerodynamic particle sizer (APS), DMA, and SP2, Zhao et al. (2020) measured absorption coefficient, scattering coefficient, size distribution and size-resolved mixing state of aerosols in eastern China, CRIs of BC containing and BC free aerosols were investigated separately based on Mie theory for sphere and core-shell structure, the corresponding CRIs were 1.67±0.67i and 1.37-1.51. Xu et al. (2016) monitored the optical properties of $PM_{1.0}$

particles during the winter heating season in Beijing, which has high concentration of black carbon aerosols, the retrieved real part of aerosol refractive index based on Mie theory is 1.40±0.06. Wang et al. (2021a) measured optical properties and size distribution of rural aerosol using a three-wavelength albedometer combined with a scanning mobility particle sizer (SMPS) and an APS, CRIs were obtained based on Mie theory by assuming aerosol as homogeneous spheres, and the fractions of preset four compositions were further clarified using volume mixing rule.

Experiment studies focus on the ambient aerosols, the inherent microphysical parameters such as morphology and mixing structures of typical black carbon and dust aerosols cannot be considered effectively, which can be explored through numerical simulation and theoretical analysis. It can be explored through numerical simulation and theoretical analysis. Wang et al. (2021c) apply a new electron-microscope-to-BC-simulation (EMBS) tool to produce shape models for BC optical calculation through DDA. The results show that the mixed structure and morphology of BC particles have a

significant effect on its radiation absorption capacity. Fierce et al. (2016) used the particle-resolved model PartMC-MOSAIC to simulate diversity in per-particle composition for populations of BC-containing particles. The results show that the composition diversity of black carbon particles significantly affects the absorption properties predicted by the model. Pang et al. (2022) developed a novel image recognition technology to automatically identify fractal dimension individuals from microscope images. Research indicated that these methods could effectively describe the fractal morphology of soot particles.

This provides an important scientific basis and methodological support for simulating individual soot models and observing the aging process of soot particles in the atmosphere. Wang et al. (2023) build a unified theoretical framework to describe the complex mixture state of black carbon and other components in the atmosphere. Research showed that the direct radiative forcing of black carbon (DRFBC) calculated using the new scheme showed significant reductions in all four selected regions: Europe, North America, South America, and Asia. Zhang et al. (2022) used HAADF-STEM and cryo-TEM

to study the behavior of black carbon aerosols during the liquid-liquid phase separation (LLPS) process and its impact on radiative absorption. They revealed that, under relative humidity below 88%, most secondary particles containing black carbon undergo phase separation, with black carbon particles tending to migrate from the inorganic salt core to the organic coating. This contributes to understanding the aging process of black carbon aerosols in the atmosphere and their environmental impacts. To represent the morphological characteristics of mineral dust, Zong et al. (2021) developed the

inhomogeneous super-spheroid model containing inside spheres, absorption, and scattering coefficients were used to obtain CRIs at 0.2-1.0 µm wavelengths based on Lorenz-Mie theory for homogeneous spheres, they stressed the adopted size distribution of spheres have significant effects on the CRIs. Furthermore, Kong et al. (2024) employed the inhomogeneous super-spheroid model, which consists of several separate mineral components, to simulate dust aerosol. The calculated

scattering and absorption coefficients were used to retrieve effective complex refractive indices (CRIs) based on homogeneous super-spheroid and sphere models. The results showed that the imaginary part of the CRIs can be retrieved more credibly from absorption than from the retrieval of both the real and imaginary parts. Zhang et al. (2019b) employed polydisperse core-shell models to represent internal-mixed BC aerosols coated by sulfate, scattering and absorption cross-sections were calculated and used to further retrieve optically effective CRIs based on single sphere Mie calculations, they revealed that the retrieved imaginary part of CRIs was significantly lower than those approximated by VWA method by 3 times. Furthermore, Zhang et al. (2019a) developed coated aggregates to represent aged BC aerosol. They simulated scattering and absorption properties using the multiple-sphere T-matrix method (MSTM) and obtained optically effective complex refractive indices (CRIs) through Mie theory. The results showed that the shell/core ratio, geometry, and size distribution have complicated effects on the retrieved CRIs; while the VWA and EMT methods performed well in predicting optical effective CRIs for aerosols in accumulation mode, they produced imaginary parts that were two times higher than the optical effective ones for coarse coated BC.

Most of the studies focusing on the optically effective CRIs, from both experimental and numerical perspectives, were conducted under the assumption that aerosols, especially black carbon (BC), are homogeneous or that their coatings are at least homogeneous. This assumption does not align with the realistic aging processes, which involve condensation, photochemical reactions, and hygroscopic growth. The transition from hydrophobic to hydrophilic is one of the uniqueness of BC aerosol after it is coated by inorganic salt or organics. During the hygroscopic growth of coated BC, atmospheric water is absorbed into coatings. The original coating materials, which are assumed to as liquid state in this study, will be diluted, the CRI of coatings will be changed gradually towards the CRI of water. Furthermore, the size of coated BC particle will be enlarged. Therefore, the optical properties including scattering and absorption change accordingly under the coupling effects of the altered lensing effect and the scattering of diluted coatings, which brings additional large uncertainties in radiative effects of BC aerosols under high relative humidity (RH).

On the other hand, if the variation of optically effective CRIs of BC coating materials at different RHs can be accurately retrieved based on their scattering and absorption properties, the water content in the coatings can then be calculated using mixing rules. This process is significant for understanding the water uptake speed of coating materials. Additionally, it can provide insights into the mechanisms of heterogeneous chemical reactions. To explore the possibility of acquiring water content in coatings from optical observations, the following questions are focused on in this study:

-Which observation wavelength and optical property are better to be employed to retrieve CRIs of coatings?

-How do microphysical properties such as morphology, size distribution, and aging degree affect the retrieved CRIs?

-What is the performance of optically effective CRIs in water content calculation at different relative humidity?

Theoretical analyses from numerical aspects are illustrated to answer the above three questions. Three typical morphological models are employed to represent BC coated by non-absorptive sulfate at different aging statuses (thinly-coated, partially-coated, and heavily-coated BC) and six RH values in the range 0-95% are selected. Scattering coefficient, absorption coefficient, and single scattering albedo of morphologically realistic aged BC at 532 and 1064 nm are calculated numerically

using the precise multi-sphere T-matrix method (MSTM). The optical equivalent CRIs of coatings at different RHs are retrieved based on Mie theory using optical properties of coated fractal BC as references, and the water content in coatings is calculated through the effective medium mixing rule.

## 2 Model and methodology

### 2.1 Models of coated BC

Freshly emitted bare black carbon particles are fractal clusters composed of many monomers, the fractal aggregates can be constructed using the DLA algorithm package based on the well-known scaling laws(Wozniak et al., 2012):

$$N = K_f (\frac{R_g}{a})^{D_f} , \tag{1}$$

$$R_g^2 = \frac{1}{N} \sum_{n=1}^{N} r_i^2 , \tag{2}$$

where a is monomer radius, $N$ is monomer number, $K_f$ is the fractal prefactor, $Df$ is the fractal dimension that controls the compactness of BC aggregate, $R_g$ is gyration radius which describes the spatial size of the aggregate, $r_i$ represents the distance between the $i$th monomer and the mass center of whole aggregate. These morphological parameters constrain the arrangement of each monomer in fractal aggregates. The optical especially absorption property is much less sensitive to $K_f$ than that to $D_f$, thus $K_f$ is set to constant 1.20, while $D_f$ varies in the range of 1.80-2.80 (Amin et al., 2019). To facilitate the conversion of volume fraction, the black carbon monomer is assumed to be monodispersed and the radius is set to 20 nm (Wu et al., 2016; Yin and Liu, 2010). To ensure the size of fractal aggregate models covers most observations of black carbon, monomer number $N$ ranges from 50 to 2000, and the maximum volume equivalent size reaches about 500 nm.

During the atmospheric aging process, bare black carbon is coated by materials like sulfate, resulting in inhomogeneous mixing structures (Kholghy, 2012). In this study, the coating material is assumed to be spherical. The volume fraction of BC ($V_f$) is taken to describe the mixing state of coated BC aerosols (Wu et al., 2014):

$$V_f = \frac{V_{BC}}{V_{total}} , \tag{3}$$

where $V_{BC}$ and $V_{total}$ represent the volume of black carbon and the whole coated particle, respectively. For a better representation of atmospheric black carbon aerosols at different aging states, the closed-cell model (CCM), partially-coated model (PCM), and coated-aggregated model (CAM) are selected and investigated, as shown in Fig. 1. The $V_f$ ranges from 1 to 0.05, the values of both $V_f$ and $D_f$ for different models is slightly different to related to aging states.

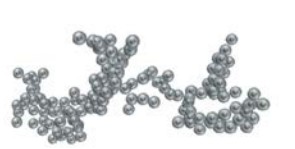 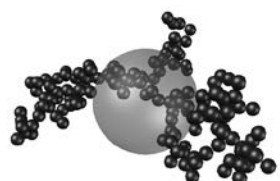 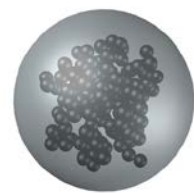

**(a) CCM with $D_f$=1.80, $V_f$=0.10  (b) PCM $D_f$=1.80, $V_f$=0.40  (c) CAM with $D_f$=2.60, $V_f$=0.10**

**Figure 1: Morphological models of coated black carbon with 150 monomers.**

## 2.2 Microphysical properties at different RHs

In this study, the bare black carbon particles are assumed to be coated by hydrophilic sulfate, and the optical properties at both 532 and 1064 nm wavelengths are investigated. Bond and Bergstrom (2006) demonstrated that the CRI of black carbon almost has no spectrum dependence within the visible and near-infrared wavelengths. The sulfate is considered to be non-absorptive, thus its imaginary part of CRI is always set to 0. CRIs of black carbon and sulfate employed are 2.26+1.26i and 1.50+0i, respectively (Zhang et al., 2019c).

Under different relative humidities in atmosphere, sulfate coatings absorb moisture and thus change both the particle size of coated BC and CRIs of coatings (Mason et al., 2015). The CRIs of sulfate solutions during the hygroscopic process can be well described by the typical two-component effective medium theory Bruggeman approximation (Luo et al., 2018a):

$$f_s \frac{\varepsilon_s - \varepsilon}{\varepsilon_s + 2\varepsilon} + f_w \frac{\varepsilon_w - \varepsilon}{\varepsilon_w + 2\varepsilon} = 0, \qquad (4)$$

$$m = \sqrt{\varepsilon}, \qquad (5)$$

where the volume fractions of sulfate and water satisfy $f_s+f_w$=1, $\varepsilon_s$ and $\varepsilon_w$ are the dielectric constants of sulfate and water respectively, $m$ and $\varepsilon$ are the effective refractive index and the effective dielectric constant for the coatings at different RHs. The variation of particle size during the hygroscopic growth of coated BC can be calculated according to the κ-Kohler theory (Zhao et al., 2022; Kuang et al., 2020):

$$RH = \frac{(D/D_i)^3 - 1}{(D/D_i)^3 - (1-\kappa)} \exp(\frac{4\partial M_w}{\rho_W RTD}), \qquad (6)$$

where $D_i$ and $D$ are the diameters of coated BC at dry and moist state respectively, $\partial$ is the surface tension of water droplet, $Mw$ and $\rho_w$ are the molar mass and density of water respectively, $R$ is the universal gas constant, $T$ is the temperature, and $\kappa$ is the hygroscopic parameter which is selected as 0.52 (Liu et al., 2014).

## 2.3 Optical simulation and CRI retrieval

Many numerical simulation methods have been developed for the optical calculation of morphological complex BC models (Zhang et al., 2018). The multi-sphere T-matrix method (MSTM), which is developed based on T-matrix theory and employs the addition theorem of vector spherical wave functions to explain the interactions between different monomers in a multi-sphere system, is efficient and accurate among all these methods (Mackowski, 2014). MSTM is suitable for sphere clusters with any morphology that contact in inside or outside form, but the limitation is that the spheres cannot overlap with each other (Mackowski and Mishchenko, 1996). With the input of CRIs for BC and coatings as well as the center coordinates and radii of spheres, the optical properties such as optical efficiency (Q), optical cross section (C), single scattering albedo (SSA) and so on for coated BC aerosol could be exactly simulated (He et al., 2015).

Aerosol scattering and absorption properties are directly related to the real and imaginary parts of CRIs respectively, thus these two parameters are commonly used for the retrieval of CRIs in laboratory and field observations (Virkkula et al., 2006). Since SSA is the ratio of scattering to extinction (sum of scattering and absorption), the performance of SSA in CRIs retrieval is examined in this study. For the optical retrieval of CRIs for coated BC, firstly, the microphysical parameters such as volume equivalent diameter of whole particle and effective CRIs of coatings under different RHs are calculated. Then, the optical properties of complex coated BC during the hygroscopic growth process are optically simulated using MSTM. Optical property look-up tables of core-shell models are constructed based on Mie theory with real part and imaginary part of CRIs ranging from 1.00-1.80 and 0-0.20 respectively. Finally, the optical equivalent CRIs of coated BC at different RHs and wavelengths are retrieved through minimum distinctions of selected optical properties between fractal model and core-shell model with the same particle size. The objective function employed for optical retrieval is as follows:

$$\chi^2 = \left( \frac{\sigma_{sca,\mathrm{MSTM}}(n,k) - \sigma_{sca,\mathrm{Mie}}(n,k)}{\sigma_{sca,\mathrm{MSTM}}(n,k)} \right)^2 + \left( \frac{\sigma_{abs,\mathrm{MSTM}}(n,k) - \sigma_{abs,\mathrm{Mie}}(n,k)}{\sigma_{abs,\mathrm{MSTM}}(n,k)} \right)^2 + \left( \frac{\mathrm{SSA}_{\mathrm{MSTM}}(n,k) - \mathrm{SSA}_{\mathrm{Mie}}(n,k)}{\mathrm{SSA}_{\mathrm{MSTM}}(n,k)} \right)^2, \tag{7}$$

where $\sigma$ is the optical properties, the subscript *sca* and *abs* are the scattering and absorption respectively, and the MSTM and Mie represent the optical properties for coated aggregate model and core-shell model.

## 3. Result and discussion

### 3.1 Performance of optical properties at different wavelengths

The optical properties for fractal BC models and core-shell models are distinct evidently to varying degrees, in some cases, the evolution of optical properties with both the real part and imaginary part of CRIs for these two models even cannot overlap, indicating that the CRI for coated BC would not be retrieved. Moreover, the retrieved real parts of CRIs can be larger than 1.50 or smaller than 1.33, and the retrieved imaginary parts of CRIs can be non-vanishing, which means that the retrieved results are physically meaningless. Figure 2 describes the retrieved real part of CRIs of coated-aggregate models

with $D_f$=2.60 and $V_f$=0.10 at different RHs, different combinations of scattering cross section (*Csca*), absorption cross
section (*Cabs*), and single scattering albedo (SSA) are selected for the CRI retrieval. The horizontal solid line and the
vertical dotted line represent the mean value and the error respectively. It can be seen that the *Csca* has the best performance
among all these optical parameters, the mean value of retrieved real part of CRIs is the closest to the preset values of
coatings, the deviation is smaller and the results are more concentrated on the mean values. However, the retrieved results
for other optical properties and their combinations are more dispersed and fluctuated. The retrieved data would diminish at
215 low humidity, there would even be no data when SSA is employed at RH=0. Therefore, only the scattering cross section is
selected for further investigation on the retrieved CRIs and water contents in coatings.

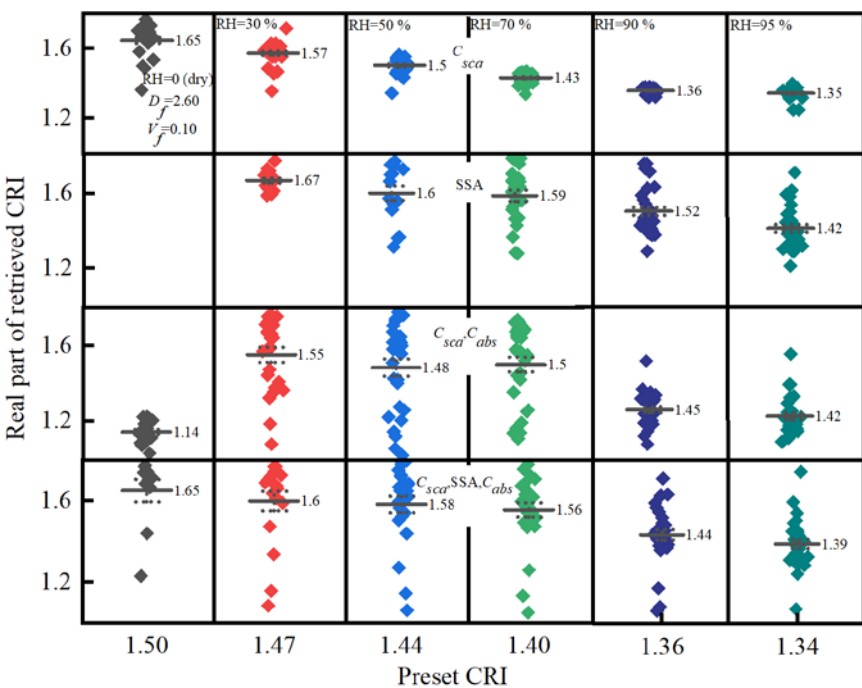

**Figure 2: Comparison of preset and retrieved real parts of CRIs based on different optical properties of coated-aggregate BC models with $D_f$=2.60 and $V_f$=0.10 at different RHs.**

Figure 3 illustrates the retrieved real part of CRIs of coated BC with closed-cell, partially-coated, and coated-aggregate
models with $V_f$=0.10 at different RHs and wavelengths. Two typical wavelengths at visible (532 nm) and near-infrared (1064
nm) spectrum are considered. The bottom and top of the boxes represent the 25th and 75th percentiles respectively, the short
lines and dots inside the boxes represent the median and mean values respectively, and the upper and lower whiskers
represent the maximum and minimum values respectively. When the atmospheric relative humidity increases from 0 to 95%,
the effective real part of CRIs for coatings after water absorption decrease gradually. The retrieved CRIs also decrease with
the RHs for different morphological models at both 532 and 1064 nm, Levoni et al. (1997) also revealed the downtrend of
the retrieved refractive indices during the hygroscopic growth. For closed-cell model, as shown in Fig. 3(a), the retrieved

CRIs at two wavelengths are underestimated under all RHs, the relative errors in retrieved values increase with RHs at 532 nm while decreasing at 1064 nm, the overall performance at 532 nm is better than that at 1064 nm, and the largest relative error at RH=95% could be about 22%. For both partially-coated and coated-aggregate models, the retrieved CRIs are overestimated during the hygroscopic process of sulfate coatings. The averaged real parts of retrieved CRIs have an obvious deviation from the preset values when the relative humidity is small, but the deviations decrease and the retrieved values are closer to the preset values with the increase of RHs, averaged relative errors for coated-aggregate with $D_f$ =2.60 and $V_f$= 0.10 decrease from 8.06% to 4.18% at 532 nm, while decrease from 8.41% to 1.53% at 1064 nm. Compared with visible wavelength, the retrieval performances at 1064 nm are much better, the distributions of retrieval results are more centralized and stable.

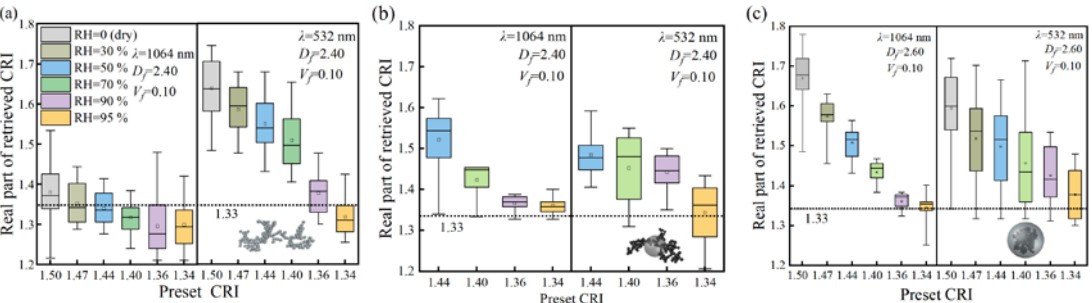

**Figure 3: Comparison of preset and retrieved real part of CRIs of coated BC aerosols at different RHs and wavelengths. (a) Closed-cell model; (b) Partially-coated model; (c) Coated-aggregate model.**

## 3.2 Influence of microphysical parameter on the retrieved CRIs

Figure 4 describes the variation of retrieved real parts of CRIs of coated-aggregate models with different BC core sizes, BC volume fractions and fractal dimensions. With the increase of particle size, the retrieved results present typical arched distribution patterns, the retrieved real parts of CRIs increase at first and then decrease, the maximum can be reached when BC core sizes are in the range of about 160-180 nm. The effects of relative humidity on CRI retrieval are evident, the arched patterns turn to slight and the fitting goodness become much better at high RHs, the retrieved real parts of CRIs are more sensitive to size distribution at low RHs. With the increase of RHs, the averaged retrieval errors decrease from 6.77% to 1.09% for BC aerosol with $D_f$ =2.80 and $V_f$=0.10. As shown in Fig. 4(b) and 4(d), the retrieved real parts of CRIs are smaller for larger BC cores with larger fractal dimensions. Furthermore, when BC volume fraction decrease from 0.10 to 0.075, the retrieved CRIs also decrease obviously.

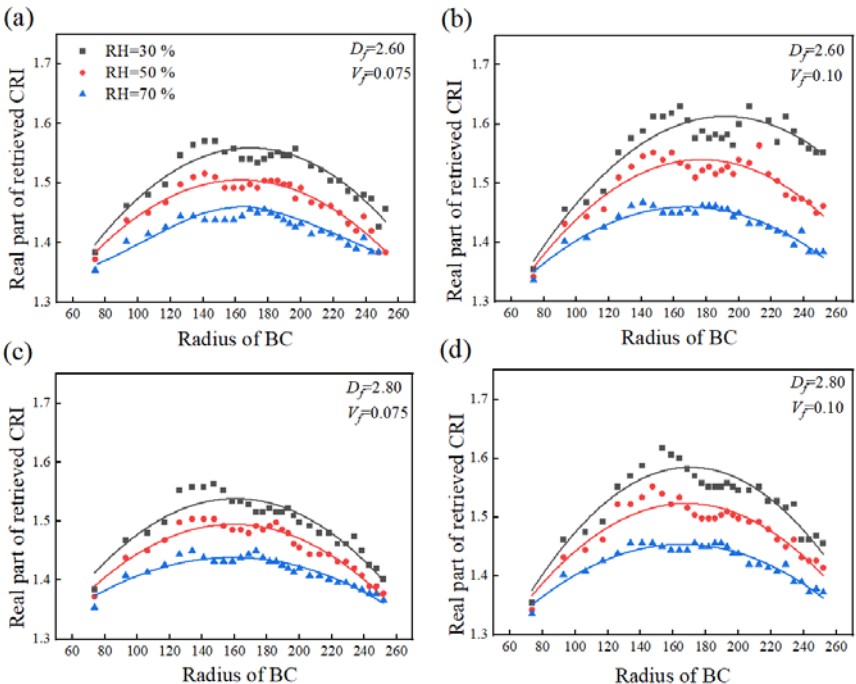

**Figure 4: Retrieved real parts of CRIs of coated-aggregate models with different fractal dimensions and BC volume fractions during the hygroscopic process. (Retrieved results are shown in points and fitted in lines)**

Figure 5 shows the variation of retrieved real parts of CRIs of partially-coated models at different RHs. Similar to coated-aggregate modes, the retrieved real parts of CRIs also increase at first and then decrease, but the maximum values are significantly affected by relative humidity. At high RHs, the sensitivity and fluctuation of CRIs with particle size are more inconspicuous. The averaged retrieval errors decrease from 4.87% to 0.93% for BC aerosol with $D_f$ =2.40 and $V_f$=0.05 under different RHs. With the BC volume fraction $V_f$ enlarged from 0.05 to 0.10, the retrieved results also increased slightly. Figure 6 shows the variation of retrieved real parts of CRIs of closed-cell models at different RHs. The retrieved results of CRIs increase with BC core radius in most cases, however, for particles with $D_f$=1.80 at dry state, the retrieved real parts of CRIs decrease at first and increase then. When RH increase from 0 to 95%, the optical equivalent real parts of CRIs decrease (Cotterell et al., 2017). During the hygroscopic growth, the mean retrieval errors decrease from 11.7% to 5.43% when the BC volume fraction and fractal dimension are 0.10 and 2.40 respectively. The retrieved refractive index would be enlarged and more and more close to the preset values. It should be noted that some of the retrieved real parts of CRIs of closed-cell models are smaller than 1.33, which is the CRI of water. (Virkkula et al., 2006) also proposed that the non-spherical morphology would lead to unreasonably low and meaningless values of effective aerosol refractive indices.

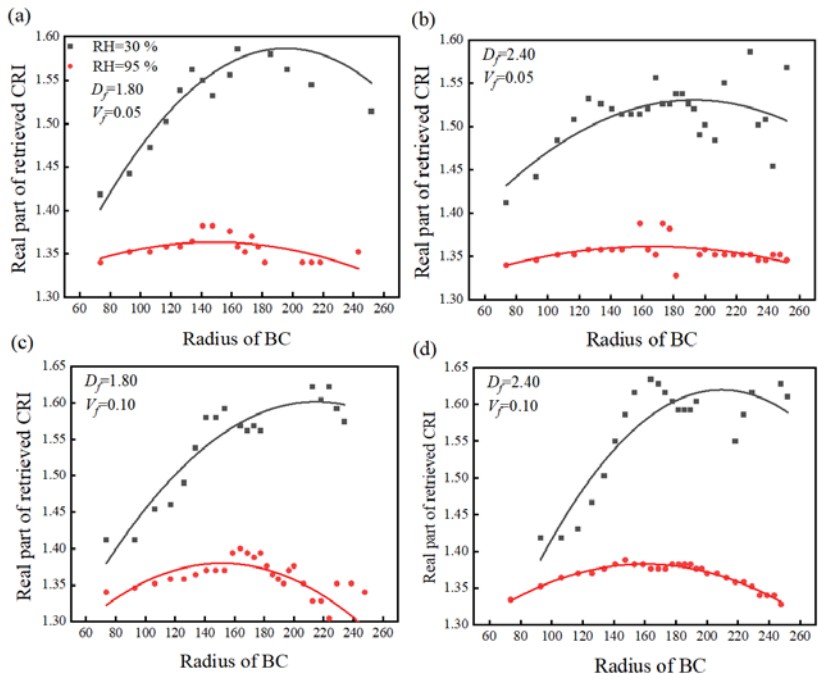

**Figure 5: Retrieved real parts of CRIs of partially-coated models with different fractal dimensions and BC volume fractions during the hygroscopic process. (Retrieved results are shown in points and fitted in lines)**

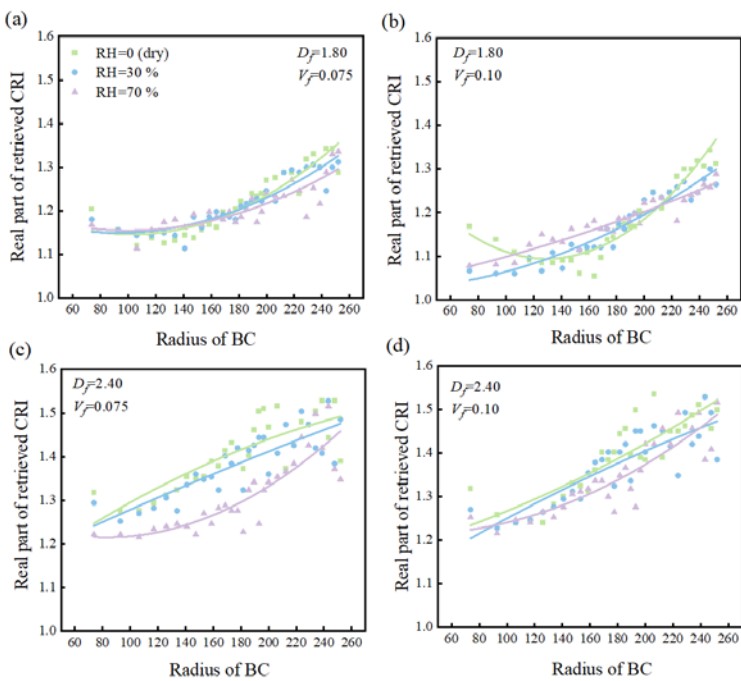

**Figure 6: Retrieved real parts of CRIs of closed-cell model with different fractal dimensions and BC volume fractions during the hygroscopic process. (Retrieved results are shown in points and fitted in lines)**

Figure 7 illustrates the comparisons of real parts of optical retrieved CRIs and the corresponding preset values during the hygroscopic process, all coated particles with different fractal dimensions, BC volume fractions, and BC core sizes are considered. The white dots in each violin plot represent the median value of retrieved real part, the areas are the probability distributions of all the data, the wider parts mean that the data is more concentrated. The optical retrieval performance is better for coated-aggregate and partially-coated models than that for closed-cell model, especially when RHs are larger than 70%. The enhanced aggregate compactness would facilitate the exact retrieval of CRIs at different RHs, the averaged retrieval error for coated-aggregate with $D_f$=2.80 could be reduced to about 6%, due to the similarity of coated-aggregate model and the core-shell model used for optical retrieval. The BC volume fractions also have some influences on the retrieval accuracy of CRIs, the retrieval errors are negligible for $V_f$=0.05 under 95% high humidity. Especially, the mean retrieval errors of CRIs for closed-cell model, partially-coated model, and coated-aggregate model reach 5.43%, 1.51%, and 1.09% respectively. With the increase of atmosphere humidity, the retrieval results are more and more reasonable, a larger amount of real part values range in 1.33-1.50, which are the refractive indices of water and sulfate respectively. Field observations conducted by Zhao et al. (2020) at areas with high concentrations of sulfate also showed that the retrieved CRIs ranged from about 1.37 to 1.51, and research by Zhang et al. (2013) showed that the real part of aerosol refractive index fluctuated around 1.50. When RHs are smaller than 30%, most of the retrieved real parts of CRIs for partially-coated and coated-aggregate models are larger than 1.50. Furthermore, the vast majority of the retrieval results (>75%) for closed-cell models are smaller than 1.33. More specifically, closed-cell models with compact structures ($D_f$ >2.40) performs better CRI retrieval than those with looser structures ($D_f$ <1.80), however, the retrieved data of the former is fewer. The fractal aggregate of BC and the non-spherical morphology of coated particles after moisture absorption leads to meaningless real parts under different RHs.

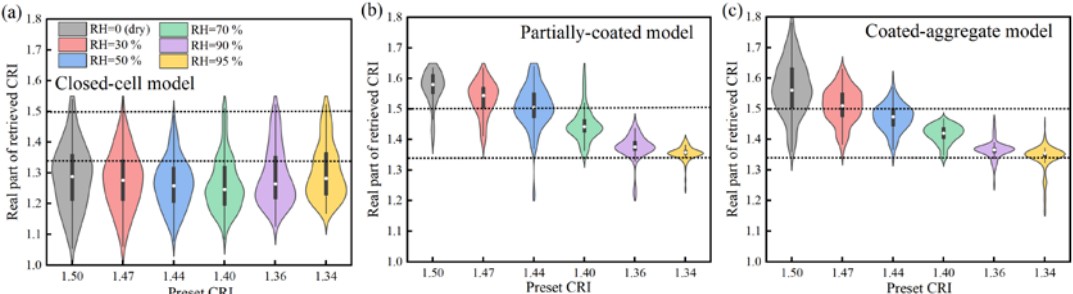

**Figure 7: Retrieved refractive indices of closed-cell model, partially-coated mode, and coated-aggregate model under different RHs.**

### 3.3 Retrieval water content in coatings from CRIs

The effective medium theory can deal well with the dielectric constant and refractive index of homogeneous mixtures of different species. Bruggeman approximation, among all the effective medium theories, is suitable for the mixture of soluble sulfate and water. Based on the retrieved optical equivalent refractive index, the water contents at different RHs can be

calculated. Figure 8 shows the comparisons of retrieved and preset water content in coatings for coated-aggregate models at different relative humidities. With RHs increasing from 30% to 95%, the retrieved water contents gradually increase, due to the enhanced water absorbing capacity (Li et al., 2024a; Bian et al., 2014). Figure 8(a-c) illustrates results for heavily aged coated-aggregate models with the same fractal dimension. When RHs are larger than 50% and BC volume fractions decrease from 0.10 to 0.05, the retrieved water contents are closer to preset values. This rule can also be seen in the 1:1 dividing lines of each subplot. However, under a low RH 30%, retrieval errors are enlarged with the decrease of $V_f$, and the averaged error is 62.68%. When $V_f$ is small, the proportion of coatings is large, therefore the retrieved water content will be more reliable. As can be seen from Fig. 8(c) and Fig. 8(f), the retrieved water contents of coated particles with larger fractal dimensions are more accurate than those with smaller $D_f$. For coated-aggregate models with large fractal dimension (2.80) and small BC volume fraction (0.05), the minimum value of retrieval errors can be about 2%. In addition, it should be noted that since the retrieved real part of CRIs can vary from 1.18 to 1.71, which are wider than the physically effective range 1.33-1.50, indicating that water contents for some coated BC particles cannot be obtained. Figure 9 illustrates the retrieved and preset water content in coatings for partially-coated models at different RHs. Three RHs (70%, 90%, and 95%) are considered for partially-coated models with $D_f$=1.80, while four RHs (50%, 70%, 90%, and 95%) are considered for the same models with $D_f$=2.40, the reason is that even through optical equivalent CRIs could be retrieved under relatively low RHs but the retrieved results are smaller than CRI of water or larger than CRI of sulfate. The obtained water contents for partially-coated models are analogous to that for coated-aggregate models, relative errors are reduced with the increase of coating amount. Nevertheless, there are more retrieved abnormal results of CRIs smaller than 1.33, and water in coatings cannot be calculated. On the other hand, the normal looking results of CRIs have large deviations from the preset corresponding values, which further result in extreme water contents. In short, the water content retrieval for BC aerosols with thick coatings, that is in a severe aging state, have the best performance.

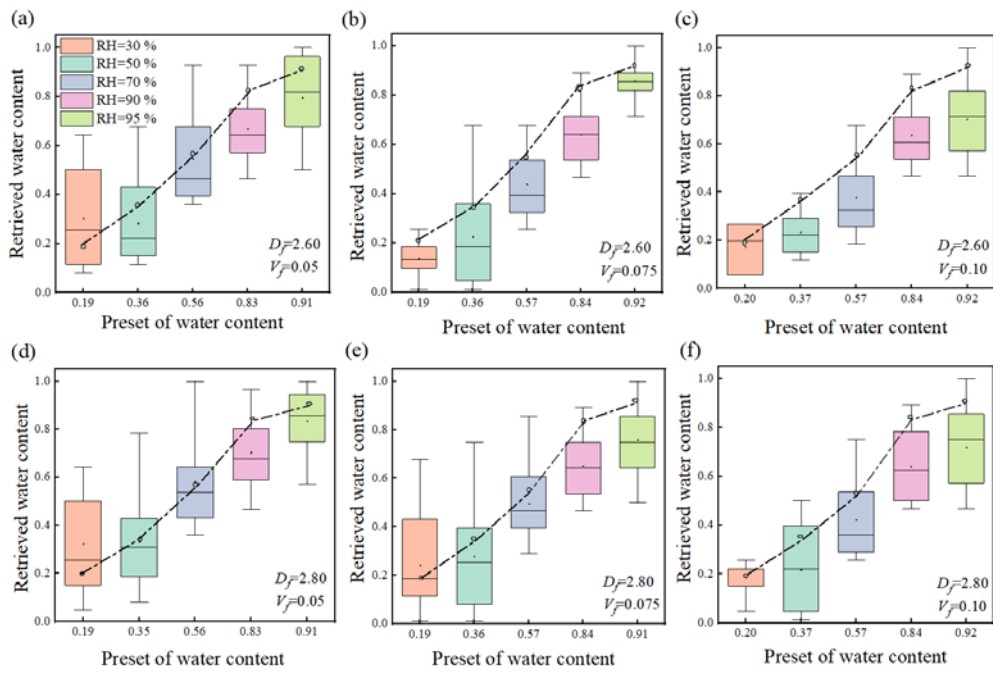

**Figure 8: Retrieved water content in coatings for coated-aggregate models with different $D_f$ and $V_f$ during the hygroscopic growth. The dotted lines are the 1:1 dividing lines.**

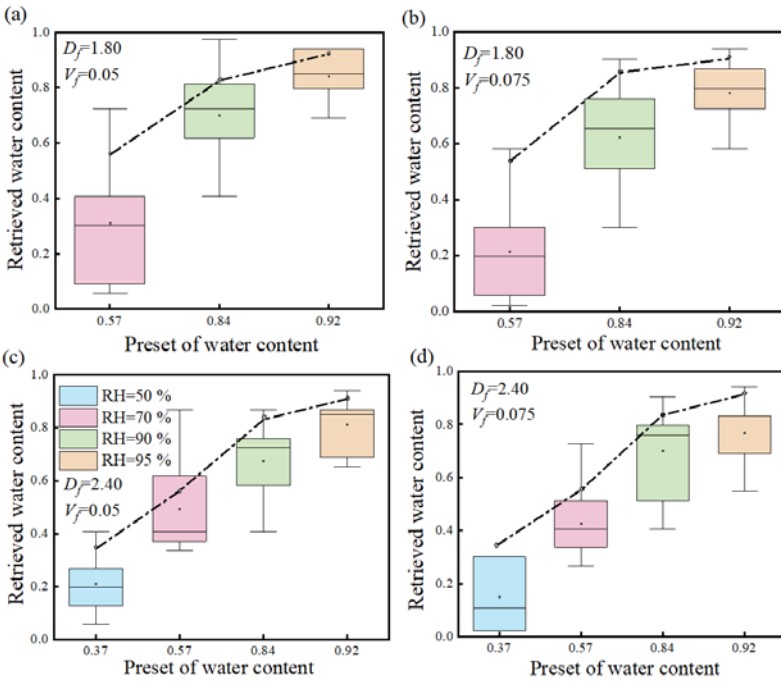

**Figure 9: Retrieved water content in coatings for partially-coated models with different $D_f$ and $V_f$ during the hygroscopic growth.**
**The dotted lines are the 1:1 dividing lines.**

## 4. Conclusion

Black carbon (BC) aerosols coated with hydrophilic materials will absorb moisture at humid environments, the quantification of water content in the coatings of BC is significant for the investigation of heterogeneous reaction and hygroscopic growth. In this study, a method to obtain water content is investigated based on the complex refractive index retrieved using optical properties. From numerical inspects, optical properties of coated BC aerosols under six RHs from 0 to 95% at both 532 and 1064 nm are simulated with the assistance of three realistic fractal models: closed-cell model, partially-coated model and coated-aggregate model. The optical equivalent complex radiative indices (CRIs) of coatings are retrieved based on core-shell Mie theory, furthermore, the water contents in coatings at different RHs are investigated theoretically through effective medium theory.

Scattering properties, among all the optical parameters and their combinations, have the best performance in retrieving CRIs of coatings of aged BC. With the RHs increase from 0 to 90%, the retrieved CRIs for closed-cell models are underestimated at both 532 and 1064 nm, and the retrieved CRIs for both partially-coated and coated-aggregate models are overestimated. The averaged relative errors for coated-aggregate models with $D_f$=2.60 and $V_f$=0.10 range from 4.18% to 8.06% at 532 nm, while relative errors range from 0.93% to 8.41% at 1064 nm. Generally, the CRI retrieval performance at 1064 nm wavelength is better. The retrieved real part of CRIs for all these three models decreases with the increased RHs, and retrieval errors also decrease. The retrieval accuracy of CRIs for coated-aggregate models are better than other two fractal models. When the RH reaches 95%, the minimum retrieval errors for closed-cell, partially-coated and coated-aggregate models are 5.43%, 1.51%, and 1.09% respectively. Fractal BC models with compact structures and small BC volume fractions performs well in the CRI retrieval. However, in certain situations such as closed-cell model with $D_f$=1.80 and $V_f$=0.10, the retrieved real part of CRIs under low humidities are meaningless, which are smaller than that of water (1.33) or larger than that of sulfate (1.50).

The water contents in the coatings of aged BC aerosols at different RHs can be effectively calculated from the optical equivalent CRIs of coatings based on Bruggeman's approximation effective medium theory. The retrieved water contents gradually increase when RHs range in 30-95%. The water content retrieval for fractal BC aerosols with smaller BC volume fractions and larger fractal dimensions are more accurate. For severely aged coated-fractal BC aerosol, the retrieval errors of water contents are the smallest, which are 2%-63%. Nevertheless, the complex morphologies of coated BC aerosols could result in unreasonable CRIs of coatings and further cause missed results of water content. This study constructs a useful method to obtain refractive index and water content of BC coatings during the hygroscopic growth process, and also highlights the possible retrieval errors caused by non-sphericity of BC aerosols when the famous core-shell Mie theory is employed for the optical retrieval.

**Fundings.** This research has been supported by the National Natural Science Foundation of China (grant no. 42305082, U23A20678), the Hebei Natural Science Foundation (grant no. D2024201001), the Science Research Project of Hebei

Education Department (grant no. BJK2024179), and the Innovation Team of Nondestructive Testing Technology and

Instrument, Hebei University (IT2023C03)

**Acknowledgments.** We particularly thank Dr. Mishchenko M. I. and Dr. Mackowski D. W. for the MSTM code. We also appreciate the support of the supercomputing center of Hebei University.

**Disclosures.** The authors declare no conflicts of interest relevant to this study.

**Data availability.** Processed data for this study are available online (https://doi.org/10.13140/RG.2.2.21765.36321).

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
