# Peer review of "Retrieval of refractive index and water content for the coating materials of aged black carbon aerosol based on optical properties: a theoretical analysis"

_EGUsphere, 2024_

## Author Comment (AC1)

**RESPONSE LETTER (EGUSPHERE-2024-1000)**

Title: Retrieval of refractive index and water content for the coating materials of aged black carbon aerosol based on optical properties: a theoretical analysis

Dear Referee:

We have revised our manuscript based on the comments. The corrections and modifications have been included in the revised manuscript and the details are listed as follows. The responses are highlighted in blue font. The changes made in the revised manuscript are marked in red font.

Comments:

This study is a theoretical analysis about the influence of water content of coatings on complex refractive index of aged black carbon. Three models are used, which are closed-cell model, partially coated model and coated-aggregate model. Varied simulations were made since different values of the most noticeable parameters involved were selected. For instance, the influence of relative humidity at different wavelengths, or particle radius was investigated. Consequently, the paper may be accepted for publication in Atmospheric Chemistry and Physics after the introduction of some minor changes.

Response:

Thanks a lot for reviewing our manuscript and all these constructive comments. We have responded to the comments point by point and modified related descriptions in the revised manuscript.

The main inconvenience that authors should correct is the lack of citations in the results section. Some references should be introduced to indicate that these results are not isolated and can be compared with those from previous analyses. This comparison is useful to discuss the results of this research.

Response:

Thank you very much for the valuable comments and suggestions! Essential investigation results in previous research have been analyzed and compared with our results, we observed that our results agreed well with previous observation and simulation investigations, the related references have been cited in the revised manuscript along with necessary descriptions as follows:

"The retrieved CRIs also decrease with the RHs for different morphological models at both 532 and 1064 nm, Levoni et al. (1997) also revealed the downtrend of the retrieved refractive indices during the hygroscopic growth."

"With the increase of atmosphere humidity, the retrieval results are more and more reasonable, a larger amount of real part values range in 1.33-1.50, which are the refractive indices of water and sulfate respectively. Field observations conducted by Zhao et al. (2020) at areas with high concentrations of sulfate also showed that the retrieved CRIs ranged from about 1.37 to 1.51, and research by Zhang et al. (2013) showed that the real part of aerosol refractive index fluctuated around 1.50. When RHs are smaller than 30%, most of the retrieved real parts of CRIs for partially-coated and coated-aggregate models are larger than 1.50."

"With RHs increasing from 30% to 95%, the retrieved water contents gradually increase, due to the enhanced water absorbing capacity (Li et al., 2024; Bian et al., 2014)."

Moreover, a link with real situations, for instance, real cases with high concentration of black carbon and high relative humidity where this theory could be applied, should be encouraged. Similar examples should reveal that presented calculations respond to real conditions.

Response:

Thank you for the constructive comments!

Our study was designed and conducted in the framework of the single scattering albedo spectrometer, for the sake of improving the retrieval accuracy of aerosol refractive index based on the optical observation from the spectrometer. With the assistance of a single scattering

albedo spectrometer, Xu et al. (2016) monitored the optical properties of $PM_{1.0}$ particles during the winter heating season in Beijing, which has high concentration of black carbon aerosols, the retrieved real part of aerosol refractive index based on Mie theory is 1.40±0.06. Zhou et al. (2020) measured the scattering, absorption, extinction coefficients and SSA at 532 nm using a humidifier cavity enhanced albedometer (H-CEA) in the laboratory, the relative humidity of H-CEA could increase from 10% to 88%. Tan et al. (2013) conducted field observation with a hygroscopic tandem differential mobility analyzer (H-TDMA) under high humidity (about 90%) at Pearl River Delta, which also has high BC concentration. In order to characterize the aerosol liquid water contents (ALWC) at North China Plain, Kuang et al. (2018) employed a three-wavelength humidified nephelometer system to measure optical properties at different RHs and showed that the measured ALWC was in good agreement with the calculated results of thermodynamic model. Therefore, according to the previous field and laboratory investigations, we conducted the numerical simulations of BC aerosols with the RH ranges from 30% to 95%.

We have added related descriptions in the revised manuscript as follows:

"Refractive index of aerosols, m=n+ki, can be determined from the scattering and absorption properties, the real part is related to the former and the imaginary part is related to the latter. Tan et al. (2013) conducted field observation with a hygroscopic tandem differential mobility analyzer (H-TDMA) under high humidity (about 90%) at Pearl River Delta, which also has a high BC concentration. In order to characterize the aerosol liquid water contents (ALWC) at North China Plain, Kuang et al. (2018) employed a three-wavelength humidified nephelometer system to measure optical properties at different RHs and they stressed that the measured ALWC was in good agreement with the calculated results of thermodynamic model. Zhou et al. (2020) measured the scattering, absorption, extinction coefficients and SSA at 532 nm using a humidifier cavity-enhanced albedometer (H-CEA) in the laboratory, the relative humidity of H-CEA could increase from 10% to 88%.

Zhao et al. (2020) measured absorption coefficient, scattering coefficient, size distribution and size-resolved mixing state of aerosols in eastern China, CRIs of BC containing and BC free aerosols were investigated separately based on Mie theory for sphere and core-shell structure, the corresponding CRIs were 1.67±0.67i and 1.37-1.51. Xu et al. (2016) monitored

the optical properties of PM1.0 particles during the winter heating season in Beijing, which has high concentration of black carbon aerosols, the retrieved real part of aerosol refractive index based on Mie theory is 1.40±0.06."

Minor remarks.

Figure-1 caption should be revised and corresponding labels should be introduced.

Response:

Thank you very much for the comments, we have modified the caption of Figure 1 in the revised manuscript as follows:

"Figure 1: Morphological models of coated black carbon with 150 monomers."

Dots after "Figure" in figure captions should be suppressed (from Figure 2 to Figure 9).

Response:

Thank you very much for pointing this out, we have suppressed the dots in the revised manuscript as follows:

"Figure 2: Comparison of preset and retrieved real parts of CRIs based on different optical properties of coated-aggregate BC models with $D_f$=2.60 and $V_f$=0.10 at different RHs."

"Figure 3: Comparison of preset and retrieved real part of CRIs of coated BC aerosols at different RHs and wavelengths. (a) Closed-cell model; (b) Partially-coated model; (c) Coated-aggregate model."

"Figure 4: Retrieved real parts of CRIs of coated-aggregate models with different fractal dimensions and BC volume fractions during the hygroscopic process. (Retrieved results are shown in points and fitted in lines)"

"Figure 5: Retrieved real parts of CRIs of partially-coated models with different fractal dimensions and BC volume fractions during the hygroscopic process. (Retrieved results are shown in points and fitted in lines)"

"Figure 6: Retrieved real parts of CRIs of closed-cell model with different fractal dimensions and BC volume fractions during the hygroscopic process. (Retrieved results are shown in points and fitted in lines)"

"Figure 7: Retrieved refractive indices of closed-cell model, partially-coated mode, and coated-aggregate model under different RHs."

"Figure 8: Retrieved water content in coatings for coated-aggregate models with different $D_f$ and $V_f$ during the hygroscopic growth. The dotted lines are the 1:1 dividing lines."

"Figure 9: Retrieved water content in coatings for partially-coated models with different $D_f$ and $V_f$ during the hygroscopic growth. The dotted lines are the 1:1 dividing lines."

Some dots after "Figure" and figure number in text should be suppressed. For instance, L. 188, L. 208, L. 220, L. 239.

Response:

Thanks for the suggestion, we have checked the full text and suppressed the dots in the revised manuscript as follows:

"Figure 3 illustrates the retrieved real part of CRIs of coated BC with closed-cell, partially-coated, and coated-aggregate models with $V_f$=0.10 at different RHs and wavelengths."

"Figure 4 describes the variation of retrieved real parts of CRIs of coated-aggregate models with different BC core sizes, BC volume fractions and fractal dimensions."

"Figure 5 shows the variation of retrieved real parts of CRIs of partially-coated models at different RHs. Similar to coated-aggregate modes, the retrieved real parts of CRIs also increase at first and then decrease, but the maximum values are significantly affected by relative humidity."

"Figure 7 illustrates the comparisons of real parts of optical retrieved CRIs and the corresponding preset values during the hygroscopic process, all coated particles with different fractal dimensions, BC volume fractions, and BC core sizes are considered."

L. 214. Introduce one space between "(b)" and "and".

Response:

Thanks a lot for the comments, we have modified this in the revised manuscript as follows:

"As shown in Fig. 4(b) and 4(d), the retrieved real parts of CRIs are smaller for larger BC cores with larger fractal dimensions."

References should follow the journal style.

Response:

Thank you very much for the comments. We have modified the references according to the style of Atmospheric Chemistry and Physics.

Furthermore, other detailed revisions are listed below.

| LOCATION | REVISED MANUSCRIPT | ORIGINAL MANUSCRIPT |
|---|---|---|
| Abstract | performs best | has the best performance |
| Introduction paragraph 2 | increased slowly | increase slowly |
| Introduction paragraph 3 | cross-sections | cross sections |
| Section 3.1, paragraph 1 | Figure 2 | Fig. 2. |
| Section 3.1, paragraph 2 | decrease | decreases |
| Section 3.3, paragraph 1 | Figure 8 | Fig. 8. |
| Section 3.3, paragraph 1 | Figure 9 | Fig. 9. |

**References**

Bian, Y. X., Zhao, C. S., Ma, N., Chen, J., and Xu, W. Y.: A study of aerosol liquid water content based on hygroscopicity measurements at high relative humidity in the North China Plain, Atmos. Chem. Phys., 14, 6417-6426, doi: 10.5194/acp-14-6417-2014, 2014.

Kuang, Y., Zhao, C. S., Zhao, G., Tao, J. C., Xu, W. Y., Ma, N., and Bian, Y. X.: A novel method for calculating ambient aerosol liquid water content based on measurements of a humidified nephelometer system, Atmos. Meas. Tech., 11, 2967-2982, doi: 10.5194/amt-11-2967-2018, 2018.

Levoni, C., Cervino, M., Guzzi, R., and Torricella, F.: Atmospheric aerosol optical properties: a database of radiative characteristics for different components and classes, Applied optics, 36, 8031-8041, doi: 10.1364/ao.36.008031, 1997.

Li, D. M., Cui, S. J., Wu, Y., Wang, J. F., and Ge, X. L.: Direct Measurement of Aerosol Liquid Water Content: A Case Study in Summer in Nanjing, China, Toxics, 12, 14, doi: 10.3390/toxics12030164, 2024.

Tan, H. B., Yin, Y., Gu, X. S., Li, F., Chan, P. W., Xu, H. B., Deng, X. J., and Wan, Q. L.: An observational study of the hygroscopic properties of aerosols over the Pearl River Delta region, Atmos. Environ., 77, 817-826, doi: 10.1016/j.atmosenv.2013.05.049, 2013.

Xu, X. Z., Zhao, W. X., Zhang, Q. L., Wang, S., Fang, B., Chen, W. D., Venables, D. S., Wang, X. F., Pu, W., Wang, X., Gao, X. M., and Zhang, W. J.: Optical properties of atmospheric fine particles near Beijing during the HOPE-J$^3$A campaign, Atmos. Chem. Phys., 16, 6421-6439, doi: 10.5194/acp-16-6421-2016, 2016.

Zhang, X. L., Huang, Y. B., Rao, R. Z., and Wang, Z. E.: Retrieval of effective complex refractive index from intensive measurements of characteristics of ambient aerosols in the boundary layer, Opt. Express, 21, 17849-17862, doi: 10.1364/oe.21.017849, 2013.

Zhao, G., Li, F., and Zhao, C. S.: Determination of the refractive index of ambient aerosols, Atmos. Environ., 240, 9, doi: 10.1016/j.atmosenv.2020.117800, 2020.

Zhou, J. C., Xu, X. Z., Zhao, W. X., Fang, B., Liu, Q. Q., Cai, Y. Q., Zhang, W. J., Venables, D. S., and Chen, W. D.: Simultaneous measurements of the relative-humidity-dependent aerosol light extinction, scattering, absorption, and single-scattering albedo with a humidified cavity-enhanced albedometer, Atmos. Meas. Tech., 13, 2623-2634, doi: 10.5194/amt-13-2623-2020, 2020.

---

## Author Comment (AC2)

**RESPONSE LETTER (EGUSPHERE-2024-1000)**

Title: Retrieval of refractive index and water content for the coating materials of aged black carbon aerosol based on optical properties: a theoretical analysis

Dear Referee:

We have revised our manuscript based on your comments. The corrections and modifications have been included in the revised manuscript and the details are listed as follows. The responses are highlighted in blue font. The changes made in the revised manuscript are marked in red font.

Comments:

This study focuses on complex refractive index of black carbon with coatings under different RHs. They retrieved the refractive index and water content for the non-absorbing coating of BC particles, and then calculated the optical properties using MSTM. This study is very basic study to retrieved the optical properties of the aged BC with different RHs. Based on my opinion, the study had one certain innovation working on the water contents of coatings, because there is limitation on water contents on coatings of BC particles and their refractive index. Basically, I like this work which provide more information for future studies.

Response:

Thanks a lot for reviewing our manuscript and for your suggestion! We have responded to the comments point by point and modified related descriptions in the revised manuscript.

Of course, we should notice there is some limitation of this study. All the study did not consider the realistic particles in ambient air. Even the authors did not cite some from other studies which has been well described the coating thickness and mixing structures. I would strongly recommend the authors should considered this issue. Moreover, the authors used lots of long sentences and subordinate clause. It is difficult for people to understand this paper well. I would ask the revisions. Certainly, I would ask the authors revised the manuscript.

Response:

Thanks a lot for reviewing our manuscript and all these constructive comments. In practical situations, the mixed state and relative humidity are quite complex. This complexity makes it challenging to find a specific scenario that connects simulations with realistic particles in ambient air. Therefore, we have taken into account a wide range of humidity and volume fractions to explore whether there are any systematic conclusions among them. This aims to provide some references for practical observational studies. Li et al. (2023) from Peking University studied the effect of black carbon content on the mass size distribution and mixing state of black carbon, finding that an increase in black carbon content leads to a greater tendency for the particles to be in an internally mixed state. Wang et al. (2017), Zhejiang University conducted a detailed characterization of the physicochemical properties of soot particles using a soot particle-aerosol mass spectrometer (SP-AMS). They stressed that soot particles with different mixing structures exhibit varying fractal dimensions, ranging from 1.80 to 2.16. Liu et al. (2020), Zhejiang University studied the microscopic morphology of particles containing refractory black carbon (rBC) using the CPMA-SP2 system. The results showed that as the coating thickness increased, the morphology of rBC-containing particles transitioned from a loose structure to a compact core-shell structure. On the whole, the mixed structures and coating thickness vary in different ambient air Thus, by analyzing microscopic images, we summarized three typical models: coated-aggregated model, closed-cell model, and partially-coated model for further study. We have cited the necessary literature for the subsequent questions, as well as in the main text.

We carefully read and reviewed the entire text, rewriting some difficult sentences and subordinate clauses. Please conduct a second review of our revised article.

Abstract

Line 18-20: the number is difficult to be understood their meanings.

Response:

Thank you for the comment! The results of refractive index and water content retrieve under different models are discussed in this study. The effects of different models and relative

humidity on the results of water content retrieve and refractive index retrieve are quite similar. For coated-aggregated model, the retrieved error for water content ranges from 2% to 63%. To clarify, we have made the following modifications to the relevant descriptions in the main text:

"The regularity of retrieved water content is similar to that of refractive index retrieve, and the water content retrieved errors range from 2% to 63% for heavily-coated BC."

Introduction:

I would like to recommend the authors revised this part carefully.

L27-28 Fresh bare BC will be coated by inorganic salts or organics during aging process such as condensation and collision in the atmosphere, and hydrophobic BC aerosol becomes hydrophilic.

This sentence should cite some references here. The BC aging process has been provided by the Li . J. Geophys. Res. 2016, 121(22): 13,784-13,798.

Response:

Thank you very much for the suggestion! In many scenarios, such as combustion in motor vehicles, a large amount of fresh, bare particles are emitted and subsequently coated with organic compounds or inorganic salts through various physical or chemical processes in the air. This phenomenon has been repeatedly confirmed by researchers in microscopic morphology analysis studies in the past few years. we have revised related description in the revised manuscript:

"Fresh bare BC will be coated by inorganic salts or organics during aging process such as condensation and collision in the atmosphere, and hydrophobic BC aerosol becomes hydrophilic(Li et al., 2016; Wang et al., 2017)."

L30-31, the sentence should cite more references here. Certainly Luo et al. is one of this study. The general conclusions should cite more references. Such as Fierce et al., Nat. Commun. 2016, 7: 12361; npj Climate and Atmospheric Science 2024, 7(1): 65. Wang et al., J. Geophys. Res. 2021, 126(10): e2021JD034620. And others.

Response:

Thank you for the meaningful comment! We have modified relevant descriptions and add the necessary references in the manuscript:

"Coating materials with different complex refractive indices produce different "lensing effect" or "sunglass effect" (Liu et al., 2021; Feng et al., 2021). In addition, the optical properties of coated BC are significantly different from those of bare BC due to the morphological changes of fractal structure, thus increasing the uncertainty of radiative effect (Luo et al., 2018; Fierce et al., 2016; Li et al., 2024; Wang et al., 2021a; Wu et al., 2017; Pang et al., 2023; Mishchenko et al., 1995)."

L60-75, the paragraph did not cover the recent studies using different methods. For example, Wang et al., Geophy. Res. Lett. 2021, 48(24): e2021GL096437. developed one EMBS method to calculate BC and their absorption; Fierce et al., Nat. Commun. 2016 developed the inhomogeneous thickness of coatings on BC particles and improved the absorption. Please make more to cover the knowledge there.

Response:

Thank you for your meaningful suggestion! The two articles you recommended are of great significance for expanding our knowledge and have made notable contributions to the development of optics, optical modeling, and the simulation of the optical properties of black carbon. Wang et al. (2021b) developed one EMBS method to calculate the optical properties of black carbon particles and explored how the differences in embedded fraction of BC particle groups in different geographic locations affect the absorption enhancement effect. Fierce et al. (2016) emphasizes the importance of considering diversity in particle composition and water uptake in determining absorption enhancement for a more accurate representation of light

absorption by BC-containing particles. We have revised the relevant descriptions and added the necessary references in the manuscript:

"which can be explored through numerical simulation and theoretical analysis. It can be explored through numerical simulation and theoretical analysis. Wang et al. (2021b) apply a new electron-microscope-to-BC-simulation (EMBS) tool to produce shape models for BC optical calculation through DDA. The results show that the mixed structure and morphology of BC particles have a significant effect on its radiation absorption capacity. Fierce et al. (2016) used the particle-resolved model PartMC-MOSAIC to simulate diversity in per-particle composition for populations of BC-containing particles. The results show that the composition diversity of black carbon particles significantly affects the absorption properties predicted by the model. Pang et al. (2022) developed a novel image recognition technology to automatically identify fractal dimension individuals from microscope images. Research indicated that these methods could effectively describe the fractal morphology of soot particles. This provides an important scientific basis and methodological support for simulating individual soot models and observing the aging process of soot particles in the atmosphere. Wang et al. (2023) build a unified theoretical framework to describe the complex mixture state of black carbon and other components in the atmosphere. Research showed that the direct radiative forcing of black carbon (DRFBC) calculated using the new scheme showed significant reductions in all four selected regions: Europe, North America, South America, and Asia. Zhang et al. (2022) used HAADF-STEM and cryo-TEM to study the behavior of black carbon aerosols during the liquid-liquid phase separation (LLPS) process and its impact on radiative absorption. They revealed that, under relative humidity below 88%, most secondary particles containing black carbon undergo phase separation, with black carbon particles tending to migrate from the inorganic salt core to the organic coating. This contributes to understanding the aging process of black carbon aerosols in the atmosphere and their environmental impacts."

L205 please display the simply model on top of the figure. Then people can know the what are these models.

Response:

Thank you for the comment! Since there was no place at the top, we searched for a suitable location at the bottom of Figure 3 in the revised manuscript to display the simple model:

[Figure]

L260 deleted the well.

Response:

Thank you for the meaningful comment! The L260 in the original manuscript has been modified as follows:

"The effective medium theory can deal with the dielectric constant and refractive index of homogeneous mixtures of different species."

L265-268, this sentence is too long. The similar sentence should be simplified.

Response:

Thank you very much for this suggestion! Changes have been made in our resubmitted manuscript.

"Figure 8(a-c) illustrates results for heavily aged coated-aggregate models with the same fractal dimension. When RHs are larger than 50% and BC volume fractions decrease from 0.10

to 0.05, the retrieved water contents are closer to preset values. This rule can also be seen in the 1:1 dividing lines of each subplot."

Similar sentences in other place of the original manuscript are simplified as follows:

"With the assistance of a self-developed cavity-enhanced albedometer, (Zhao et al., 2014; Xu et al., 2016) measured the extinction coefficient, scattering coefficient, absorption coefficient and single scattering albedo (SSA) for atmospheric aerosols at Jing-Jin-Ji Area. The effective CRI of aerosols is retrieved based on the Mie theory of homogeneous sphere by using the optical properties and volume mixing. The real part of CRIs is about 1.38 ~ 1.44, and the imaginary part is about 0.008 ~ 0.04.

they stressed the adopted size distribution of spheres have significant effects on the CRIs. Furthermore, Kong et al. (2024) employed the inhomogeneous super-spheroid model, which consists of several separate mineral components, to simulate dust aerosol. The calculated scattering and absorption coefficients were used to retrieve effective complex refractive indices (CRIs) based on homogeneous super-spheroid and sphere models. The results showed that the imaginary part of the CRIs can be retrieved more credibly from absorption than from the retrieval of both the real and imaginary parts.

Zhang et al. (2019a) developed coated aggregates to represent aged BC aerosol. They simulated scattering and absorption properties using the multiple-sphere T-matrix method (MSTM) and obtained optically effective complex refractive indices (CRIs) through Mie theory. The results showed that the shell/core ratio, geometry, and size distribution have complicated effects on the retrieved CRIs; while the VWA and EMT methods performed well in predicting optical effective CRIs for aerosols in accumulation mode, they produced imaginary parts that were two times higher than the optical effective ones for coarse coated BC.

Most of the studies focusing on the optically effective CRIs, from both experimental and numerical perspectives, were conducted under the assumption that aerosols, especially black carbon (BC), are homogeneous or that their coatings are at least homogeneous. This assumption does not align with the realistic aging processes, which involve condensation, photochemical reactions, and hygroscopic growth.

On the other hand, if the variation of optically effective CRIs of BC coating materials at different RHs can be accurately retrieved based on their scattering and absorption properties,

the water content in the coatings can then be calculated using mixing rules. This process is significant for understanding the water uptake speed of coating materials. Additionally, it can provide insights into the mechanisms of heterogeneous chemical reactions."

Finally, I would ask what kinds of condition should be considered the water contents. In realistic, could you provide how much difference between water and non-water in coatings influence optical properties of aged particles? If the author could make such comparison, this could be important for the potential readers.

Response:

Thanks a lot for your valuable comments! During the actual measurement process, there is indeed a significant amount of humidity. Li et al. (2021) conducted experiments to study the behavior of particles collected in different environments (forest and city) under varying relative humidity (RH) conditions. They stressed that particles began to grow at an RH of 50% and transitioned to a liquid state when the RH increased to 84% or 83%. Zhang et al. (2023) studied the collapse of particle soot structure and changes in coating composition during long-distance transport. The results showed that when the relative humidity (RH) is between 60% and 90%, it is conducive to forming secondary aerosol coatings on soot particles and facilitates the transition of soot from a partially coated state to an embedded state. Zhao et al. (2018) calculated the aerosol asymmetry factor (g) using the humidifying turbidity meter system. The g value of dry aerosols varies between 0.54 and 0.67, while at an environmental relative humidity of 90%, the g value increases significantly by 1.2 times. The varying water content of combustible materials, such as wood and straw, can result in different humidity in the environment. We agree that the difference between water and non-water in coatings effects the optical properties of aged particles. The following figures show the comparison of different models under different hygroscopic conditions. It can be seen from the figure that the relative humidity has a significant positive influence on the optical characteristics. When the relative humidity reaches 95%, the scattering cross section increases by 10.84, 6.35 and 7.14 times, respectively, compared with different models when the relative humidity is 0.

[Figure]

Furthermore, other detailed revisions are listed below.

| LOCATION | REVISED MANUSCRIPT | ORIGINAL MANUSCRIPT |
|---|---|---|
| Abstract | performs best | has the best performance |
| Introduction paragraph 2 | increased slowly | increase slowly |
| Introduction paragraph 3 | cross-sections | cross sections |
| Section 3.1, paragraph 1 | Figure 2 | Fig. 2. |
| Section 3.1, paragraph 2 | decrease | decreases |
| Section 3.3, paragraph 1 | Figure 8 | Fig. 8. |
| Section 3.3, paragraph 1 | Figure 9 | Fig. 9. |

References

Feng, X., Wang, J. D., Teng, S. W., Xu, X. F., Zhu, B., Wang, J. P., Zhu, X. J., Yurkin, M. A., and Liu, C.: Can light absorption of black carbon still be enhanced by mixing with absorbing materials?, Atmos. Environ., 253, 8, doi: 10.1016/j.atmosenv.2021.118358, 2021.

Fierce, L., Bond, T. C., Bauer, S. E., Mena, F., and Riemer, N.: Black carbon absorption at the global scale is affected by particle-scale diversity in composition, Nat. Commun., 7, 8, doi: 10.1038/ncomms12361, 2016.

Kong, S. Y., Wang, Z., and Bi, L.: Uncertainties in laboratory-measured shortwave refractive indices of mineral dust aerosols and derived optical properties: a theoretical assessment, Atmos. Chem. Phys., 24, 6911-6935, doi: 10.5194/acp-24-6911-2024, 2024.

Li, F., Luo, B., Zhai, M. M., Liu, L., Zhao, G., Xu, H. B., Deng, T., Deng, X. J., Tan, H. B., Kuang, Y., and Zhao, J.: Black carbon content of traffic emissions significantly impacts black carbon mass size distributions and mixing states, Atmos. Chem. Phys., 23, 6545-6558, doi: 10.5194/acp-23-6545-2023, 2023.

Li, W. J., Riemer, N., Xu, L., Wang, Y. Y., Adachi, K., Shi, Z. B., Zhang, D. Z., Zheng, Z. H., and Laskin, A.: Microphysical properties of atmospheric soot and organic particles: measurements, modeling, and impacts, npj Clim. Atmos. Sci., 7, 14, doi: 10.1038/s41612-024-00610-8, 2024.

Li, W. J., Teng, X. M., Chen, X. Y., Liu, L., Xu, L., Zhang, J., Wang, Y. Y., Zhang, Y., and Shi, Z. B.: Organic Coating Reduces Hygroscopic Growth of Phase-Separated Aerosol Particles, Environ. Sci. Technol., 55, 16339-16346, doi: 10.1021/acs.est.1c05901, 2021.

Li, W. J., Sun, J. X., Xu, L., Shi, Z. B., Riemer, N., Sun, Y. L., Fu, P. Q., Zhang, J. C., Lin, Y. T., Wang, X. F., Shao, L. Y., Chen, J. M., Zhang, X. Y., Wang, Z. F., and Wang, W. X.: A conceptual framework for mixing structures in individual aerosol particles, J. Geophys. Res.-Atmos., 121, 13784-13798, doi: 10.1002/2016jd025252, 2016.

Liu, H., Pan, X. L., Liu, D. T., Liu, X. Y., Chen, X. S., Tian, Y., Sun, Y. L., Fu, P. Q., and Wang, Z. F.: Mixing characteristics of refractory black carbon aerosols at an urban site in Beijing, Atmos. Chem. Phys., 20, 5771-5785, doi: 10.5194/acp-20-5771-2020, 2020.

Liu, L., Zhang, J., Zhang, Y. X., Wang, Y. Y., Xu, L., Yuan, Q., Liu, D. T., Sun, Y. L., Fu, P. Q., Shi, Z. B., and Li, W. J.: Persistent residential burning-related primary organic particles during wintertime hazes in North China: insights into their aging and optical changes, Atmos. Chem. Phys., 21, 2251-2265, doi: 10.5194/acp-21-2251-2021, 2021.

Luo, J., Zhang, Y. M., Wang, F., and Zhang, Q. X.: Effects of brown coatings on the absorption enhancement of black carbon: a numerical investigation, Atmos. Chem. Phys., 18, 16897-16914, doi: 10.5194/acp-18-16897-2018, 2018.

Mishchenko, M., Lacis, A., Carlson, B., and Travis, L.: Nonsphericity of dust-like tropospheric aerosols: Implications for aerosol remote sensing and climate modeling, Geophys. Res. Lett., 22, 1077-1080, doi: 10.1029/95gl00798, 1995.

Pang, Y. E., Chen, M. H., Wang, Y. Y., Chen, X. Y., Teng, X. M., Kong, S. F., Zheng, Z. H., and Li, W. J.: Morphology and Fractal Dimension of Size-Resolved Soot Particles Emitted From Combustion Sources, J. Geophys. Res.-Atmos., 128, 13, doi: 10.1029/2022jd037711, 2023.

Pang, Y. E., Wang, Y. Y., Wang, Z. C., Zhang, Y. X., Liu, L., Kong, S. F., Liu, F. S., Shi, Z. B., and Li, W. J.: Quantifying the Fractal Dimension and Morphology of Individual Atmospheric Soot Aggregates, J. Geophys. Res.-Atmos., 127, 11, doi: 10.1029/2021jd036055, 2022.

Wang, J. D., Wang, J. P., Cai, R. L., Liu, C., Jiang, J. K., Nie, W., Wang, J. B., Moteki, N., Zaveri, R. A., Huang, X., Ma, N., Chen, G. Z., Wang, Z. L., Jin, Y. Z., Cai, J., Zhang, Y. X., Chi, X. G., Holanda, B. A., Xing, J., Liu, T. Y., Qi, X. M., Wang, Q. Q., Pöhlker, C., Su, H., Cheng, Y. F., Wang, S. X., Hao, J. M., Andreae, M. O., and Ding, A. J.: Unified theoretical framework for black carbon mixing state allows greater accuracy of climate effect estimation, Nat. Commun., 14, 8, doi: 10.1038/s41467-023-38330-x, 2023.

Wang, Y. Y., Pang, Y. E., Huang, J., Bi, L., Che, H. Z., Zhang, X. Y., and Li, W. J.: Constructing Shapes and Mixing Structures of Black Carbon Particles With Applications to Optical Calculations, J. Geophys. Res.-Atmos., 126, 15, doi: 10.1029/2021jd034620, 2021a.

Wang, Y. Y., Liu, F. S., He, C. L., Bi, L., Cheng, T. H., Wang, Z. L., Zhang, H., Zhang, X. Y., Shi, Z. B., and Li,

W. J.: Fractal Dimensions and Mixing Structures of Soot Particles during Atmospheric Processing, Environ. Sci. Technol. Lett., 4, 487-493, doi: 10.1021/acs.estlett.7b00418, 2017.

Wang, Y. Y., Li, W. J., Huang, J., Liu, L., Pang, Y. E., He, C. L., Liu, F. S., Liu, D. T., Bi, L., Zhang, X. Y., and Shi, Z. B.: Nonlinear Enhancement of Radiative Absorption by Black Carbon in Response to Particle Mixing Structure, Geophys. Res. Lett., 48, 10, doi: 10.1029/2021gl096437, 2021b.

Wu, Y., Cheng, T. H., Zheng, L. J., and Chen, H.: Sensitivity of mixing states on optical properties of fresh secondary organic carbon aerosols, J. Quant. Spectrosc. Radiat. Transf., 195, 147-155, doi: 10.1016/j.jqsrt.2017.01.013, 2017.

Zhang, J., Li, W. J., Wang, Y. Y., Teng, X. M., Zhang, Y. X., Xu, L., Yuan, Q., Wu, G. F., Niu, H. Y., and Shao, L. Y.: Structural Collapse and Coating Composition Changes of Soot Particles During Long-Range Transport, J. Geophys. Res.-Atmos., 128, 13, doi: 10.1029/2023jd038871, 2023.

Zhang, J., Wang, Y. Y., Teng, X. M., Liu, L., Xu, Y. S., Ren, L. H., Shi, Z. B., Zhang, Y., Jiang, J. K., Liu, D. T., Hu, M., Shao, L. Y., Chen, J. M., Martin, S. T., Zhang, X. Y., and Li, W. J.: Liquid-liquid phase separation reduces radiative absorption by aged black carbon aerosols, Communications Earth & Environment, 3, 9, doi: 10.1038/s43247-022-00462-1, 2022.

Zhao, G., Zhao, C. S., Kuang, Y., Bian, Y. X., Tao, J. C., Shen, C. Y., and Yu, Y. L.: Calculating the aerosol asymmetry factor based on measurements from the humidified nephelometer system, Atmos. Chem. Phys., 18, 9049-9060, doi: 10.5194/acp-18-9049-2018, 2018.

---

## Author Comment (AC3)

**RESPONSE LETTER (EGUSPHERE-2024-1000)**

Title: Retrieval of refractive index and water content for the coating materials of aged black carbon aerosol based on optical properties: a theoretical analysis

Dear Referee:

We have revised our manuscript based on your comments. The corrections and modifications have been included in the revised manuscript and the details are listed as follows. The responses are highlighted in blue font. The changes made in the revised manuscript are marked in red font.

Comments:

This study aims to compute the optical properties of morphologically realistic soot--water mixtures. The authors have done a nice job of summarizing the results of MSTM calculations. The modeled particles represent 3 different morphologies: pure soot, partly coated, and fully encapsulated. (These are termed CCM, PCM, and CAM). I would like to suggest that the authors modify their PCM to represent partial compaction.

Multiple studies have observed that partial compaction occurs for partially coated soot. These studies have recently been reviewed in Corbin, Modini, and Gysel (https://doi.org/10.1080/02786826.2022.2137385) and in Sipkens and Corbin (https:./doi.org/10.1016/i.carbon.2024.1 19197). The former study describes the physics of this restructuring and the latter study quantifies the relationship of coating to compaction. l recommend that the authors modify their PCM model to reflect the fact that almost all laboratory and field studies have observed that partly coated soot is partly collapsed.

Response:

Thank you for your attention to our article and all these constructive comments. The compaction of soot is primarily influenced by internal mixing mechanisms such as condensation and evaporation processes. The degree of partial compaction can be assessed by considering the number of primary particles, effective density, and changes in coating volume. Soot will be partially compacted before being completely encapsulated. Regardless of the

coating material, soot aggregates generally become fully compacted following a 5-fold increase in volume (Corbin et al., 2023; Sipkens and Corbin, 2024). Soot-aggregate restructuring may occur when individual particles change from a smaller fractal dimension to a larger one due to surface tension. It can also involve different particles experiencing collision or restructuring, or be influenced by the coating material, where some primary soot particles are entered into the coating during the compaction process. In addition, the higher fractal dimensions of partially-coated models in this paper can lead to compaction to a certain extent. The relationship of coating to compaction may be more complex in real atmospheric environments. More observations and additional modeling focused on restructuring under different coating components are needed. We have added the necessary changes in the manuscript:

"and hydrophobic BC aerosol becomes hydrophilic. Zhang et al. (2023) studied the collapse of particle soot structure and changes in coating composition during long-distance transport. The results showed that when the relative humidity (RH) is between 60% and 90%, it is conducive to forming secondary aerosol coatings on soot particles and facilitates the transition of soot from a partially coated state to an embedded state. Soot-aggregate restructuring is a complex phenomenon influenced by various factors, including the physical and chemical properties of the coating materials and the environmental conditions to which the soot is exposed. Soot compaction is mainly influenced by internal mixing mechanisms. Soot is partially compacted before full coating and typically becomes fully compacted after a fivefold increase in volume, regardless of the coating material (Corbin et al., 2023; Sipkens and Corbin, 2024)."

Furthermore, other detailed revisions are listed below.

| LOCATION | REVISED MANUSCRIPT | ORIGINAL MANUSCRIPT |
|---|---|---|
| Abstract | performs best | has the best performance |
| Introduction paragraph 2 | increased slowly | increase slowly |

| Introduction paragraph 3 | cross-sections | cross sections |
|---|---|---|
| Section 3.1, paragraph 1 | Figure 2 | Fig. 2. |
| Section 3.1, paragraph 2 | decrease | decreases |
| Section 3.3, paragraph 1 | Figure 8 | Fig. 8. |
| | Figure 9 | Fig. 9. |

References

Corbin, J. C., Modini, R. L., and Gysel-Beer, M.: Mechanisms of soot-aggregate restructuring and compaction, Aerosol Sci. Technol., 57, 89-111, doi: 10.1080/02786826.2022.2137385, 2023.

Sipkens, T. A. and Corbin, J. C.: Effective density and packing of compacted soot aggregates, Carbon, 226, 10, doi: 10.1016/j.carbon.2024.119197, 2024.

Zhang, J., Li, W. J., Wang, Y. Y., Teng, X. M., Zhang, Y. X., Xu, L., Yuan, Q., Wu, G. F., Niu, H. Y., and Shao, L. Y.: Structural Collapse and Coating Composition Changes of Soot Particles During Long-Range Transport, J. Geophys. Res.-Atmos., 128, 13, doi: 10.1029/2023jd038871, 2023.